# Total Fat Gravimetric Method Workflow in Portuguese Olives Using Closed-Vessel Microwave-Assisted Extraction (MAE)

**DOI:** 10.3390/foods10102364

**Published:** 2021-10-04

**Authors:** João Gonçalo Lourenço, Daniel Ettlin, Inês Carrero Cardoso, Jesus M. Rodilla

**Affiliations:** 1Departamento de Química, Universidade da Beira Interior, Rua Marquês d’Ávila e Bolama, 6200-001 Covilhã, Portugal; joao.goncalo.lourenco@ubi.pt; 2Unicam Sistemas Analíticos, Alameda António Sérgio, 1495-132 Algés, Portugal; daniel.ettlin@thermounicam.pt; 3ISEC–Instituto Superior de Engenharia de Coimbra, Rua Pedro Nunes, 3030-199 Coimbra, Portugal; a21270885@isec.pt; 4Departamento de Química, Universidade da Beira Interior, FibEnTech-Materiais Fibrosos e Tecnologias Ambientais, Rua Marquês d’Ávila e Bolama, 6200-001 Covilhã, Portugal

**Keywords:** microwave-assisted extraction, total fat, soxhlet, NIR, olives

## Abstract

A simple and rapid method for the quantitation of total fat in olive samples is designed, evaluated, and presented. This method is based on an innovative closed-vessel microwave-assisted extraction (MAE) technique. A method was designed for olives, and some figures of merits were evaluated: limit of detection (LOD), limit of quantification (LOQ) and expanded uncertainty (U). The data obtained in these experiences show that the workflow of the MAE method in a closed container is statistically equivalent to the other two methods, showing in this case better performance indicators (LOD = 0.02%, LOQ = 0.06%, and U = 15%). In addition, it is also demonstrated that the complete MAE method workflow allows the determination of total fat in a maximum of 12 analyses simultaneously for about 100 min in each run, which is the capacity of the rotor. This is a much better productivity when compared to the traditional Soxhlet-based method. Considering the sample workflow, the closed-vessel MAE method greatly simplifies sample handling, therefore minimizing sample loss during sample preparation and reducing analysis time. When MAE is compared to NIR-based methods, the advantage comes from there being no need for any type of calibration in the sample matrix. The MAE method itself can be used to determine the reference value for NIR calibration purposes. The results obtained for CRM using MAE were equivalent to the ones shown on the certificate.

## 1. Introduction

Presently, consumers are concerned about fat ingestion. As fat affects the texture and taste of food, and considering the change of food habits, the potential risk of consuming excess calories from fat is considerable. These changes of habits have created a large concern within the community surrounding the alleged relationship between excessive calorie ingestion, obesity, and health problems. Fats are primarily defined as the tri-esters of fatty acids and glycerol, and thus are commonly called triglycerides. Solid triglycerides are commonly referred to as fat, and liquid triglycerides are commonly called oils. Lipids, on the other hand, include all the “fatty” materials in a food (i.e., generally those materials dissolved in a fat-solubilizing solvent). This includes sterols, mono, di, and triglycerides, phospholipids, glycolipids, free fatty acids, fat soluble vitamins, etc. To clarify some of the terminologies, the Food and Drug Administration, in Chapter 21 of CFR 101.9-NUTRITION LABELING OF FOOD, defines fat as the sum of the fatty acids in the food, regardless of source, that are expressed as triglyceride equivalents. These fatty acids may be present as free fatty acids, mono, di, and triglycerides, phospholipids, glycolipids, or sterol lipids. Individual fatty acids are classified according to their degree of unsaturation. These classifications include saturated, monounsaturated, polyunsaturated, and trans fatty acids [1].

Some terminology can be applied indistinctly to the diversity of substances known as fat, fatty acids, or lipids. For that purpose, to defend consumer and public health, the use of nutrition labels is a regular practice in many countries. This may enable consumers to choose healthier food habits. There is a key parameter on the labels that assesses the quantity of fat in food, and particularly on olives, described as “total fat”. EU Regulation No. 1169/2011 of the European Parliament approved new rules on food labeling, making labels clearer and more readable and allowing consumers an easier choice of product they wish to purchase. All members states of the European Union must follow this regulation of food labeling since December 2016. The meaning of total fat, according to this regulation, is “total lipids, including phospholipids” [2]. 

Consumers recognize the advantages of nutritional labeling and see them as important when choosing foods. The proper parameter evaluation is of utmost importance for necessary routine adjustments by industry players, bearing in mind the purpose of the quality assurance of food as well as regulation compliance. 

Several methods have been described for total fat determination. Some of them are described by the Association of Official Analytical Chemists (AOAC) [3], the International Organization for Standardization (ISO) [4,5,6], Portuguese-related norms [7], ultrasound extraction methods [8], and the Folch method [9].

Some of the methods used as reference mention the type of matrix that correspond to them. Among many of different approaches, and according to the matrix analyzed, one of the common and dominant techniques in the industry is the ISO extraction method, based on the Weibull Stoldt strategy [4]. This type of method as a general overview is based on, first, primary digestion using hydrochloric acid and water. The filtered residue is extracted at another stage with an adequate nonpolar solvent in a Soxhlet extractor for several hours. After the extraction, the solvent is evaporated and dried until it reaches a constant weight. The quantitation of total fat is determined by weighing the residue left. 

The AOAC methodologies are widely used. In succinct terms, this method also uses an acid digestion and posterior extraction using a nonpolar solvent (like n-hexane, diethyl ether, and/or petroleum ether) in a Mojonnier flask, or a similar device. Both ISO and AOAC methods are based primarily on a sequential method that has a mandatory first step based on digestion and a posterior Soxhlet extraction step. Any of these methods can be used as an arbitration or reference method to determine the total fat parameter in food samples to create the nutritional label. Many countries refer to arbitration methods based on any of the methodologies cited. Nevertheless, in academia, the Folch method [9] is commonly used in research and studies, especially in the field of nutrition, biochemical, and agriculture research [3]. 

When considering the case of meat products and gravimetric classic total fat determinations, the methodologies are based on the Weibull Stoldt method and involve a workflow comprising two main sequential steps. In the first one there is sample saponification by means of acid hydrolysis, employed to solubilize some of the protein, eventually digesting other compounds, but mainly to liberate fat before extraction. The second step is a Soxhlet extraction with a nonpolar solvent. 

The objective of this study is to evaluate and compare the capabilities of a new workflow with a microwave-assisted extraction method for olive samples to determine their total fat content. We would like to show the advantages of such an approach compared to the reference methods. Therefore, results are compared with the ones obtained by the traditional gravimetric Soxhlet-based methodology. We also used near-infrared spectroscopy, as it can also be used in some service laboratories and industry labs for quality control or process control. 

## 2. Soxhlet Method

The Soxhlet method is quite a well-known methodology in the laboratory and a traditional method. It dates to the 19th century, and the original purpose was the determination of fat in milk [10]. The sample is deposited inside a receptacle, which is continuously soaked in a fresh solvent condensate (the solvent used in extraction) of a distillation flask. The solvent goes up until it reaches a certain level and overflows through a siphon, therefore carrying the solvent and extracted material into the distillation flask. This operation is repeated several times, over a certain time period, usually hours, to ensure complete extraction. In operational terms, this is a continuous and discreet technique, as the solvent is permanently recirculated through the sample cartridge.

Extraction using Soxhlet methodology has some advantages. One of the appointed ones is that the sample is placed several times and almost continuously in contact with the solvent. In this situation, there is a displacement of the equilibrium transfer for better extraction. It is also convenient that the sample cartridges may avoid any type of sample filtration. Additionally, investment is quite low compared with other techniques, but especially, its overall simplicity makes Soxhlet one of the most used techniques in the routine laboratory [11]. The methodology has been part of routine methods for many laboratories for many years. 

Compared to other and more recent techniques of solid-sample preparation extraction, the Soxhlet method has some known drawbacks, including long extraction times. As the extraction is matrix-dependent, there is some difficulty in confirming that the process is quantitatively finished, and a proper validation of the time needed for extraction is compulsory. We should not forget that the use of large quantities of solvents gives rise to some concern relating to environmental issues. The inherent difficulty of stirring the sample and making better contact with the solvent can lead to inaccurate results. In general, even though we recognize some simplicity, it is a workflow that involves quite a lot of processing steps, and that reason is enough to lead to human error. That is the main reason there have been some attempts to automate it, not always removing its recognized disadvantages [12]. Some attempts to improve the performance by speeding up and introducing some automatization to both the extraction and hydrolysis steps, were reviewed by Ullah et al. [12]. 

## 3. Closed-Vessel Microwave-Assisted Extraction Method

Microwave-assisted extraction is quite a recent method for extracting soluble products in a particular fluid from a wide range of materials. In those methodologies, microwave energy is applied by some means to the system [13]. 

This approach may have many advantages over similar technologies, the most important one being that extractions are faster, with better yield and therefore more effective. Also worth considering is the lower energy consumption and less environmental waste as it uses less organic and toxic solvent volume [14].

The MAE extraction process is based inherently on the capability of a given matrix to absorb microwave energy. The amount of absorption varies with the chemical nature of the species that are irradiated with microwaves [15]. One of the parameters that is used to analyze how large the energy absorption could be is the dielectric constant. When applying microwave energy to a system, a higher dielectric constant results in more energy absorption and, consequently, there will be a temperature increase of the media (see Table 1).

Water is a strong microwave absorber, which is an immediate consequence of its high dielectric constant due to its particular molecule configuration. Therefore, in the case that water is present in the matrix or in the system, it will absorb a significant fraction of the irradiated microwave energy. Sometimes even the sole moisture of a sample can be enough to promote energy absorption at the same time as the extraction is being carried out. On a liquid-phase extraction using MAE, it is recommended that the matrix be immersed with solvents that have a better capability of dissolving the target compounds. However, despite its adequacy for the dissolution of the sample, it may present a small dielectric constant. Consequently, these solvents are almost transparent to microwaves, and therefore the heating of the system is very slow. To overcome this negative effect, it is common to add co-solvents with a higher dielectric constant, which will promote a temperature increase on their own [16]. The solvent mix chosen for the MAE method is adapted by taking into account the compromise of the ability to dissolve the desired analytes in the matrix and their transparency relative to the microwaves. Cyclohexane was chosen over other solvents, because it could fulfil some of the properties for a good and quantitative fat extraction. It has a similar polarity (0.006 relative polarity to hexane, 0.009 relative polarity) [17]. It also has a similar dielectric constant (2.02) compared with hexane [18]. Additionally, a higher boiling point would help the manipulation of solvents and avoid losing any by evaporation, yielding better quantitative results. As low polarity is very important for fat extraction, this characteristic will make microwave absorption from this solvent quite inefficient. The existence in the solvent media of water will contribute to microwave absorption and therefore a more effective heating of the reaction vessel content.
foods-10-02364-t001_Table 1Table 1Dielectric constant of some solvents [19].SolventDielectric Constant
Hexane
1.9
Isopropanol
18.3
Water
80.4

MAE is generally a technique that allows the extraction of compounds from a given matrix over a shorter period of time compared to conventional techniques [14]. There is one important additional consideration on the total fat extraction workflow method proposed: hydrolysis and extraction are carried out in just one single step and inside a pressure-controlled closed vessel. Inside the closed vessel there is the simultaneous presence of an acidic solution of water to produce hydrolysis, and a nonpolar solvent for fat extraction. Because of the elevated water content of the acidic aqueous phase and the consequent promotion of high microwave energy absorption, there is a fast temperature increase of the media. To facilitate the interaction of solvents and sample on both hydrolysis and extraction processes, there is a magnetic stirrer that mixes the system thoroughly and continuously. The heating of the solvents inside the closed vessel consequently prompts the rise of pressure. That will increase the vapor pressure of the system, and will lead to the direct potential advantage that temperature can be maintained above the boiling points of both media. Therefore, the extractions and hydrolysis will happen at a higher temperature compared with atmospheric pressure ones. Temperature and therefore the pressure can be controlled and programmed, managing the microwave power applied to the system. The possibility of having a higher temperature inside the vessels has a dramatic effect on the kinetics of the hydrolysis and the extraction speed. It promotes a faster extraction and hydrolysis speed, with excellent recovery rates over other conventional Soxhlet methods. 

## 4. NIR Method

Near-infrared (NIR) spectroscopy offers a rapid, objective, non-destructive, and simultaneous analysis of several traits at much lower cost per sample compared with the common reference laboratory methods. Because of this, the food industry has increased its interest in NIR spectroscopy implementation for processes control by at-line, on-line, and in-line measurements. Olive is a quite a difficult sample to analyze using NIR spectroscopy because of the wide variability of types and its heterogeneity [19]. Although all organic components have absorption bands in the NIR region, minerals can only be detected if they are in organic complexes, or indirectly due to changes produced in the amount of hydrogen bonding by minerals, which affects the water spectrum) [20].

NIR spectroscopy, since its appearance on the commercial market in the 1970s, has been a compelling technique in the development of chemometrics. NIR spectroscopy measures molecular vibrations in the near-infrared range and is related to the absorption of electromagnetic radiation in the wavelength range of 780–2500 nm. NIR spectra are originated by basic molecular vibrations of a variety of combinations of tones and taps, practically throughout the infrared region. Absorption bands are quite broad, so this type of spectrum is quite difficult to interpret using conventional approaches of molecular absorption techniques. This process has some advantages such as good accuracy and the ability to measure high-dispersion multi-stage dispersion matrices.

When applied in food area as bands of NIR spectra, bands are understood to be distributed from overlapping absorptions. Bands correspond to overtones of the combinations of the vibrational modes involving chemical bonds C-H, O-H, and N-H [21].

In principle, water, fat, protein, and carbohydrate concentrations can be determined using classical absorption spectroscopy. However, what happens in most food samples are changes in spectra related to physical properties, such as particle size. Due to these aspects, NIR spectroscopy becomes a secondary method since it requires a calibration based on a reference method. Therefore, as a result of the physics of diffuse transmittance, reflectance and complexity of the spectra leads to a calibration that is commonly performed by multivariate mathematics (chemometrics) [21].

Currently, NIR spectroscopy is used for compositional, functional, and sensorial analysis of food composition, intermediate processes in production, and the final product. As in many workflows, sample preparation is minimal, and the time of analysis is quite short (between 15 and 90 s). It is observed as a quick and simple method for routine measurements, specially taking into account the ability to measure several constituents simultaneously [21].

One of the major disadvantages of the NIR analysis is that the user must rely on less accurate calibrations, extensive and sophisticated mathematic models, and very complex and continuous validations. It should be pointed out that all calibration samples need to be quantified using the reference method for each parameter measured. [20]. It should be noted that there is an ISO method that refers to type of technology, providing that the gaps are considered [22]. 

## 5. Material, Methods, and Procedure

### 5.1. Samples

The olives to be tested were collected in Alentejo, a southern subregion of Portugal. For this study, four different samples of olive were used. The olives were ground, and an amount was weighed for analysis. All samples were homogenized adequately and subdivided for the four different workflows. Each workflow refers to each sample.

A certified reference material (CRM) was also acquired. 

### 5.2. Reagents 

Cyclohexane ≥99.5% (Reag. Ph. Eur), for analysis, AC, ISO, Panreac. CAS number 110-82-7 Molecular Weight 84.16, C_6_H_12_; Sulfuric Acid 95–97% (Reag. Ph. Eur) for analytical reagent, Riedel–de Haën, CAS number 7664-93-9 Molecular Weight 98.08, H_2_SO_4_.

## 6. Microwave-Assisted Extraction Method (MAE)

The method is based on a simultaneous acid hydrolysis and organic solvent extraction in a pressure-regulated closed Teflon vessel. A similar method based on that already tested and developed by Rodilla, Lourenço, Ettlin et al. was used [23]. This methodology was reviewed as potentially universal for total fat measurement. MAE was carried out using an ETHOS-X advanced microwave system with a 12-sample rotor (Milestone, Bergamo, Italy), where each vessel can withstand up to 30 bar of pressure. The temperature of the extraction is measured continuously and controlled on one of the extraction vessels by a fiber optic temperature control, with a precision of 0.1 °C. 

The homogenized sample of olive (using from 0.5 g to 2.0 g in different runs) was weighed in a Teflon vessel of 100 mL capacity, and carefully wetted and mixed with a volume of sulfuric acid 25% (10.0 mL). After that, cyclohexane (25.0 mL) is added to the same Teflon vessel, and accurately weighed. A Teflon-coated stirring bar is added carefully inside the vessel and all of them are closed and positioned inside the rotor of the microwave unit. 

The MAE was run using a temperature program with a ramp designed to reach 125 °C in 4 min and then holding that temperature for 40 more minutes. During the whole program, magnetic stirring (at 120 rpm) was applied, to enhance extraction and mixing. The maximum microwave power applied was 1200 W, and it was automatically adjusted by software to follow the above-mentioned temperature program (Figure 1).

In this case, acid hydrolysis by means of the acid solution is occurring simultaneously to extraction, therefore saving several steps compared with the Soxhlet conventional chemical extraction processes. 

After irradiation, the closed Teflon vessels were allowed to cool automatically for approximately 30 min by forced air flow until ambient temperature is reached inside the microwave oven. 

After this, the vessels were carefully opened. The organic phase containing the total fat constituent was on the top layer of the vessel. An aliquot of 10 mL of this organic phase was carefully removed and transferred to an aluminum cup and accurately weighed. The cups were positioned inside an evaporation rotor (Milestone^®^ Rar-400 rotor) with Weflon^®^ made supports, where heating and common vacuum conditions apply for each cup. The heat temperature program is designed to sustain 105 °C for 30 min. The temperature was controlled by an infrared controller. The controller monitored each cup temperature, and power was adjusted automatically according to the measured temperature, and to follow the designed program. As the cyclohexane is almost transparent to microwaves, the Weflon^®^ supports in contact with the aluminum cups promote organic solvent gentle heating, compensating for the lack of energy absorption from the extracted samples. When applying power and temperature, each aluminum cap was under a vacuum, which facilitates faster evaporation of the solvent, in a gentle and controlled manner, minimizing potential oxidation of fat. After the heating and vacuum program, the cups were removed from the rotor, allowed to cool, and weighed to achieve the result of total fat gravimetrically. As the cups are made of aluminum, ambient temperature is reached very fast, allowing almost immediate weighing of the cups. 

A brief schematic comparison of the traditional workflow using Soxhlet and the new proposed microwave-assisted extraction method can be observed in Figure 2.

## 7. Results and Discussion 

For the verification and validation of results, the following steps were followed as detailed in Figure 3:

### 7.1. Equivalence Statistical Test

It has already been verified that the MAE workflow had statistical equivalence with a validated or referential method. As shown by Rodilla et al. [23], a statistical model was used for other matrices. In particular, MAE results were compared against Soxhlet results. Applying the *t*-test, an experimental value (T_exp_) that was less than the one in the T tabled (T_crit_) for statistic equivalence verification was obtained.

### 7.2. Method Optimization 

At this point, an understanding is sought of the experimental conditions that can be varied with a view to obtaining better results.

It may seem logical that a mass of 2.00 g should be used, considering the potential total fat present in the sample, and a sufficient mass amount to be weighed would yield a good recovery. To confirm the proper mass amount, several different weights were tested, from 0.50 g and 2.00 g. During the tests, it was found that for 2.00 g mass, a suspension of the post-hydrolysis sample in microwave vessels was formed. That was not so evident for lower masses. Therefore, it was decided to choose a mass of 1.00 g, which presented the best binomial behavior experimental vs. statistical data. The best combination of data was acquired from a good manipulation of the sample after hydrolysis, and a proper statistical acceptance. The latter is defined as CV< 10%, following a current rule of laboratory analysis, which can be found in the Nordtest approach [24,25].

For masses of 0.75 g and 0.50 g, the established criterion was not observed, as shown in Table 1, Table 2, Table 3, Table 4 and Table 5, on the Table 4 and Table 5 below.

Different types of olives were used (samples identified as 87, 88, 89, and 90), and run in duplicate and triplicate in different runs numbered 1, 2, 3, and 4. Runs 1 and 2 were analyzed in triplicate and Runs 3 and 4 in duplicate. For easy understanding, Runs 1, 2, 3, and 4 correspond to the olive samples 87, 88, 89, and 90, respectively. For each mass sample, the number of tests could be different, according to the criterion of CV < 10%. 

The experimental results are shown in Table 2, Table 3, Table 4 and Table 5: 

**Table 2 foods-10-02364-t002:** Results for 1.00 g.

№ of Tests	Sample	Average	Standard Deviation	CV
** *n* ** ** = 12**	1	17.46%	1.57%	9.00%
** *n* ** ** = 12**	2	20.08%	1.36%	6.78%
** *n* ** ** = 8**	3	19.43%	0.81%	4.15%
** *n* ** ** = 8**	4	23.72%	0.78%	3.27%

**Table 3 foods-10-02364-t003:** Results for 2.00 g.

№ of Tests	Sample	Average	Standard Deviation	CV
** *n* ** ** = 3**	1	19.13%	1.58%	8.25%
** *n* ** ** = 3**	2	22.08%	0.79%	3.59%
** *n* ** ** = 2**	3	22.04%	0.81%	3.69%
** *n* ** ** = 2**	4	23.32%	0.18%	0.76%

**Table 4 foods-10-02364-t004:** Results for 0.75 g.

№ of Tests	Sample	Average	Standard Deviation	CV
** *n* ** ** = 6**	1	17.27%	2.24%	12.96%
** *n* ** ** = 6**	2	21.96%	2.29%	10.45%
** *n* ** ** = 4**	3	21.86%	1.24%	5.69%
** *n* ** ** = 4**	4	22.17%	1.62%	7.31%

**Table 5 foods-10-02364-t005:** Results for 0.50 g.

№ of Tests	Sample	Average	Standard Deviation	CV
** *n* ** ** = 3**	1	20.02%	4.32%	21.56%
** *n* ** ** = 3**	2	22.61%	2.63%	11.62%
** *n* ** ** = 2**	3	24.66%	4.09%	16.57%
** *n* ** ** = 2**	4	27.51%	1.46%	5.29%

For better observation of the data, a control chart was built for each sample. For the purposes of quality control in the assay, control charts were drawn up for each sample, aiming to represent the dispersion of values between several repeated assays (two or more) over time and under conditions of intermediate precision. 

The control charts contain a set of lines that will allow the operator to know whether the process is under control, as can be seen in the examples presented (Figure 4, Figure 5, Figure 6 and Figure 7). The central line represents the average of the measurements.
♦The high limit alert line (HLA) corresponds to a flag that can alert the operator that the values are outsider the adequacy of the methodology. Usually, this is obtained from the average line plus 2 s, s being the standard deviation of the readings.♦The low limit alert line (LLA) represents in a similar way as the LAS, but in this case, it is defined statistically from the average less a 2 s value.♦Highest limit control flag (HLC) corresponds to the average plus 3 s, which is the preferred control situation.♦Lowest limit of control (LLC) is defined statistically from the average less a 3 s value.

Below, you can see the control charts for each sample.

## 8. Validation Study

Finally, validation was performed, with the estimation of the analytical figures of merit for the limit of detection (LOD) and limit of quantitation (LOQ), and the precision, accuracy, combined uncertainty, and expanded uncertainty.

### 8.1. LOD and LOQ

For the determination of the analytical values LOD and LOQ, 2 blanks in different runs were included, making a total of 18 blank tests, using the same reagents and procedure (sulfuric acid 25% + cyclohexane). In Table 6 the results can be observed.

The Grubbs test was used to choose the proper results, to verify the presence of extreme values in sample observations. For the 18 samples and a 0.05% confident interval, the G value is 2.475. We concluded that there is one aberrant value, and we eliminated it from calculations.

In this case, the proper result is accepted if G_exp_ < G_crit_. G_crit_ is tabled in the follow reference, from the respective degrees of freedom [26].

The calculation of the LOD and LOQ was determined using the equations listed below, with $x_{0}$ as obtained measurement and $S_{0}$ as standard deviation as referred to references [25,26].
(1)$$\mathrm{LOD}=x_{0}+3S_{0}$$
(2)$$\mathrm{LOQ}=x_{0}+10S_{0}$$

The value obtained for LOD was 0.00185 g of total fat, and for the LOQ a value of 0.00583 g. That corresponds to a maximum limit of 10 g/10 mL of sample, referred as 0.02% and 0.06%, taking into account the limitations of the vessel volume and the sample volume.

### 8.2. Precision Uncertainty

The calculation of precision used the relative amplitude between duplicates of the samples approach, in intermediate precision conditions. With that purpose, the duplicates of Samples 3 and 4 [24,25] were used.

From the equation of reference
*u_p_* = *Sp_i_* × 100(3)
and
(4)$$Sp_{i}=\frac{Relative\;Amplitude\;Average}{1.128*}$$
*1.128 is a constant associated with intermediate precision determination when relative amplitude is used. This value is mentioned in ISO 11352:2012 (2012). Water quality estimation of measurement uncertainty is based on validation and quality control.

A value of 5.14% is observed for precision uncertainty.

### 8.3. Bias Uncertainty

To obtain the value of the accuracy/bias, the same type of analysis was developed using the same experimental workflow described. To measure bias, a certified reference material (CRM) was used. The CRM was supplied by “Gabinete de Servicios para la Calidad”. The results are described in [27].

To determine bias, the determination and analysis against the value of the CRM was used. The bias uncertainty obtained was of 5.71% and calculated from the equation [25,26]:(5)$$U_{bias}=\sqrt{bias^{2}+{\left(\frac{Standard\;deviation}{\sqrt{24}\times Average}\right)}^{2}{\left(\frac{\frac{U}{2}}{Real\;Value}\right)}^{2}}\times 100$$

### 8.4. Combined and Expanded Uncertainty

For this calculation, we have used the following equation [24,25]:(6)$$U_{c}=\sqrt{u_{R_{W}}^{2}+u_{b}^{2}}$$

The value obtained was 7.68%.

Then, from equation
*U* = *k u_c_* = 2 *u_c_*
(7)
we have obtained a value of 15% for the expanded uncertainty (*U*).

To verify and observe the results of the run on a graphical format, the already mentioned control chart was built, as seen in Figure 8, with respective data in Table 7.

With that chart, it is possible to deduce that none of the values are above the flag lines or the maximum allowed ones. Therefore, we can foresee the repeatability for this assay in subsequent determinations, and the low probability of systematic errors that may appear. This shows the feasibility of the methodology to be used for any total fat determination. We can anticipate that similar results will be observed in matrices with a similar total fat content, as per the chemical nature of the extraction itself and the workflow.

## 9. Conclusions

From the exhibited results, we can conclude that the MAE is an excellent alternative for total fat determination of olive oils, against the traditional Soxhlet methodology or equivalent. The values observed in the CRM are of good correspondence with the experimental ones. For reference purposes, and as an official method, Soxhlet is usually preferred. However, the closed-vessel MAE workflow methodology was found to be a very good alternative for total fat determination in olives. This type of workflow seems to have an easier manipulation process and be less prone to error, though some practice by the operator may be necessary. Furthermore, closed-vessel MAE proves to have a big advantage in speed of determination, productivity, and more effective extractions. As the solvent use is less and has the possibility of recovery, it is also an environmentally friendly method.

MAE shows an enormous potential for total fat determination, being a method that allows the performance of up to 12 samples simultaneously and one that can be used in other types of samples of a nature other than olives. It is obvious to see than due to the nature of the hydrolysis and extraction process, samples with similar expected values of total fat will have a similar performance to the one we have seen in this article. We also foresee that for difficult samples with low fat content, it may provide a better alternative due its precision and extraction yield.

## Figures and Tables

**Figure 1 foods-10-02364-f001:**
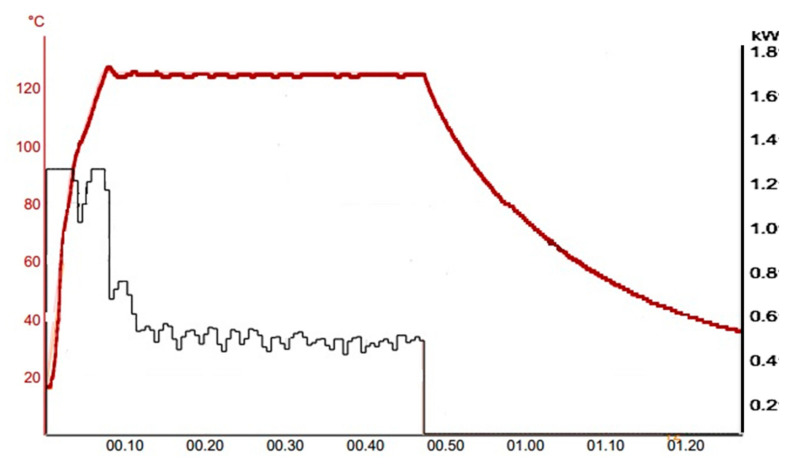
Temperature/power vs. time extraction program plot.

**Figure 2 foods-10-02364-f002:**
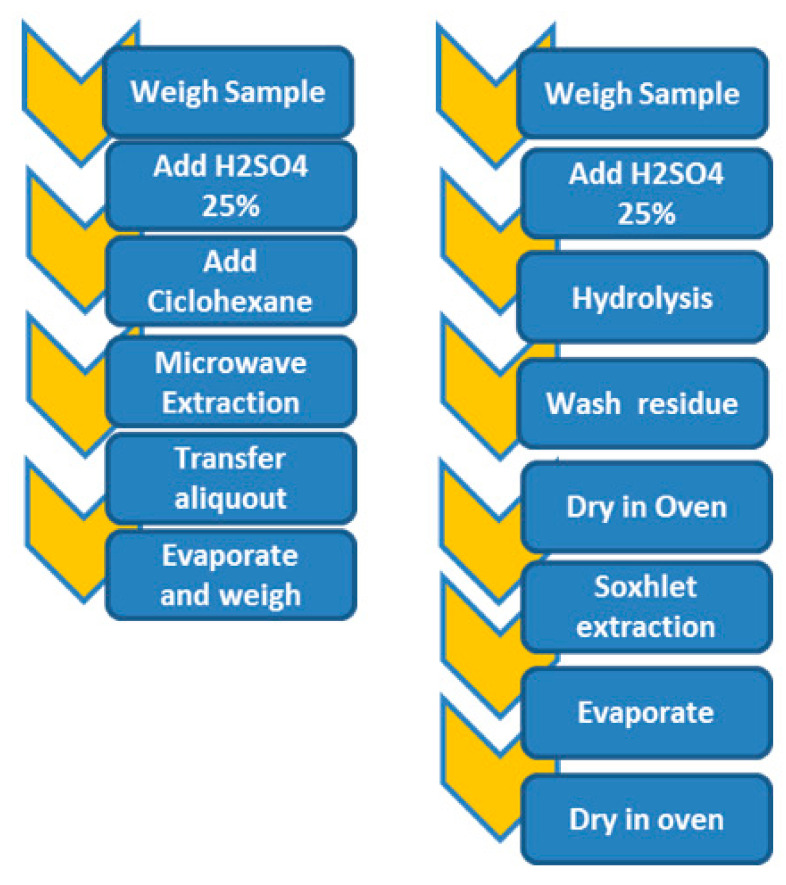
Closed-vessel microwave extraction and Soxhlet-based method workflow comparison.

**Figure 3 foods-10-02364-f003:**
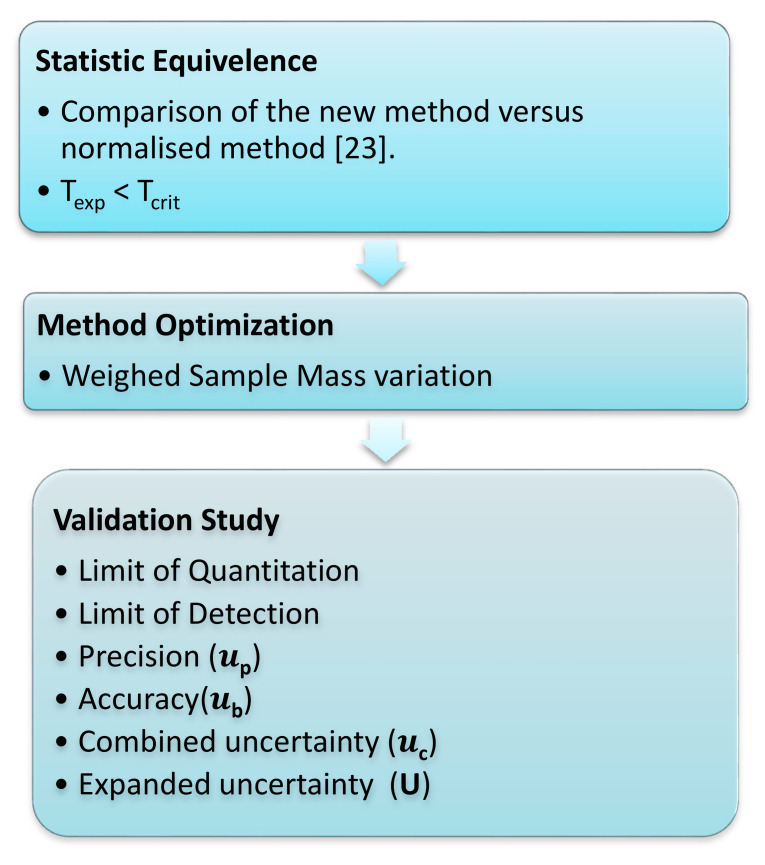
Validation workflow.

**Figure 4 foods-10-02364-f004:**
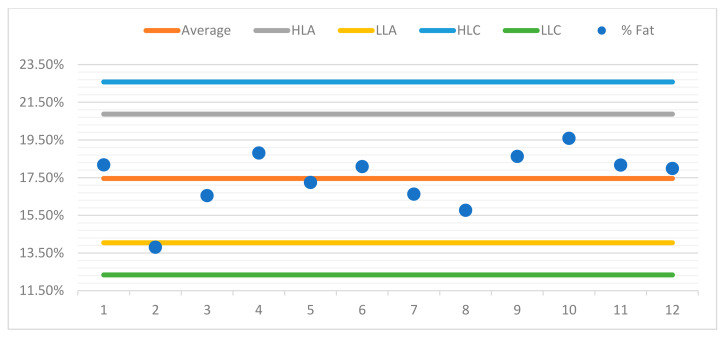
Control chart sample 1.

**Figure 5 foods-10-02364-f005:**
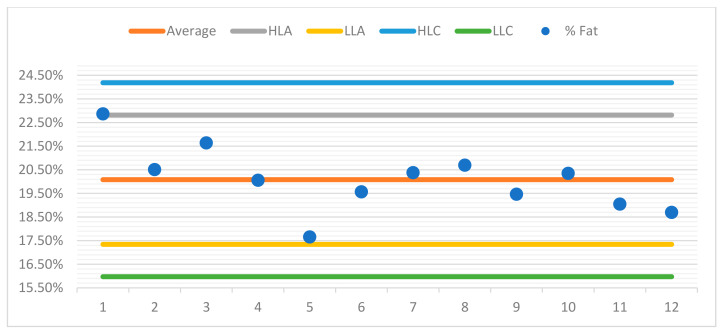
Control chart sample 2.

**Figure 6 foods-10-02364-f006:**
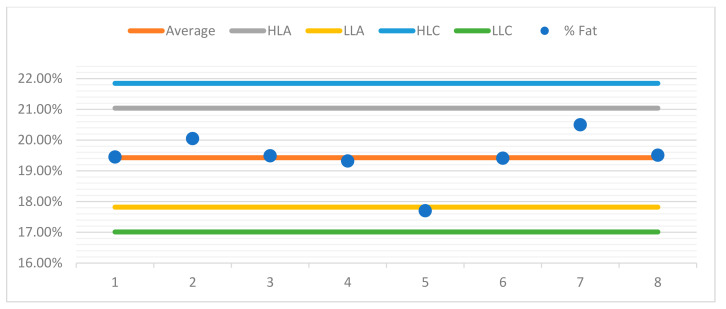
Control chart sample 3.

**Figure 7 foods-10-02364-f007:**
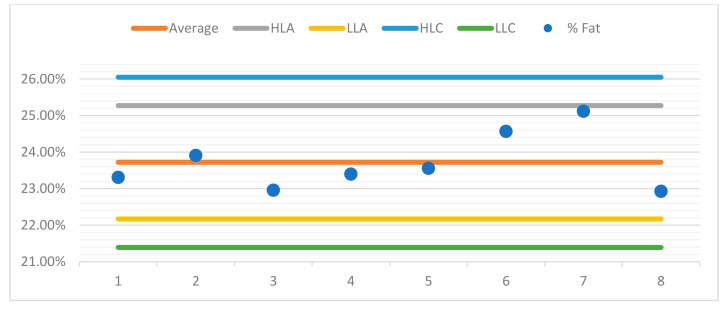
Control chart sample 4.

**Figure 8 foods-10-02364-f008:**
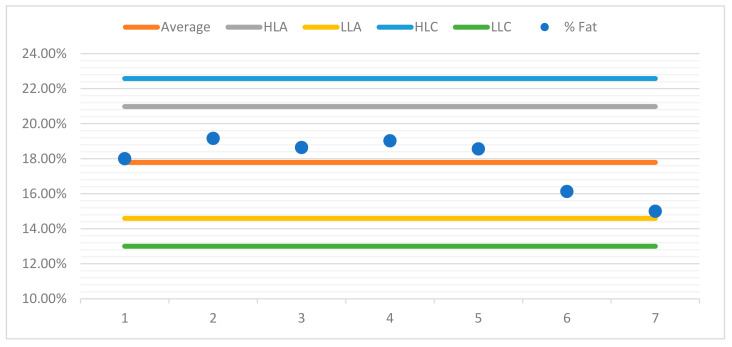
Control chart for CRM.

**Table 6 foods-10-02364-t006:** Results for blank runs and G_exp_ value.

Blank	G exp
**0.0001**	0.246
**0.0018**	0.415
**−0.0001**	0.324
**0.0007**	0.013
**0.0003**	0.168
**0.0005**	0.091
**−0.0006**	0.518
**−0.0002**	0.363
**0.0005**	0.091
**0.0006**	0.052
**0.0108**	**3.914**
**0**	0.285
**0**	0.285
**0**	0.285
**−0.0004**	0.441
**−0.0001**	0.324
**−0.0003**	0.402
**−0.0004**	0.441

Bod letter: This measurement was out of limits, so it was decided to use bold letter.

**Table 7 foods-10-02364-t007:** Results for CRM 1.00 g.

№ of Tests	Sample	Average	Standard Deviation	CV	Real Value	U
** *n* ** ** = 7**	CRM	17.79%	1.60%	3.53%	17.68%	0.24%

## Data Availability

Data is contained within the article.

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
