# Peer review of "Total Fat Gravimetric Method Workflow in Portuguese Olives Using Closed-Vessel Microwave-Assisted Extraction (MAE)"

_foods, 2021, doi:10.3390/foods10102364_

Round 1

Reviewer 1 Report

The topic is of great interest. Unfortunately, graphs, tables appear in the paper without being properly explained in the writing or even without any reference to them in the text. There is no references to support the equations used to calculate the  figures of merit.

The following list try to highlight the main drawbacks that I have found to understand this work (P: page; L: line)

1.- P.9 L 325. Equivalence Statistical Test:  in this paragraph the procedure is not clear (do the authors apply Soxhlet method to some samples?). It seems that the equivalence test is passed, does not it?

2.- P.9 L332: method Optimisation. I understand that samples 87, 88, 89, 89, 90 correspond to runs 1, 2, 3, 4 respectively. But, then, the column “Nº of Tests”  should be the same in tables 2,3,4,5, but it is not.

As a matter of fact only Table 3 and 5 seem right attending to L. 338 -339 (“runs number 1 and 2 were analysed in triplicate and runs 3 and 4 in duplicate”)  which seem to say that 10 measurements should appear in each Table.

3.- Figures 4, 5, 6, 7 correspond to samples 1, 2, 3 4  in Table 2?.  If so, average line in the graph is not coherent with the average in Table 2.  Besides this, control charts are represented over time (L. 374) but the authors do not say that measurements are taken throughout time.

4.-P.12 L.401 Validation study. These are the “figures of merit”, is there a relevant reference from which  the authors take the corresponding definitions that apply later (Lines:  421, 424, 435, 454, 458 461) ?

5.- P.12 and 13 L. 406-418  As 2 blanks are included per run, Table 6 should have 20 values instead of 18 ?  Grubbs test only detect if the maximum or minimum value is an outlier under normal distribution of the measurements, is this the goal  here? What do the authors mean with “interval of 0.05% “ in L.414, what is the role of Table 9 in this study?

6.- From P.15 L.476 to P.19 L. 486 tables and graphs appear  without any explanation appear. What is their meaning? What do they appear for? What the reader should conclude after reading and studying each of these tables?

Minor comments:

1.-P.10 L.364 some reference for criterion CV<10%?

2.- L 380, 383, 385, 386  LAS, LAI , LCS;LISàHLA, LLA, ….

3.- P. 13 L.421, 423, what   x0 and S0 are.

4.- L.426 de sample?

5.- L. 440 and L.449 Why these references are not at the end of the paper?

6.- P.14 L454 translate to English

7.- L.468 Figure 81?

8.- L. 474 matrixes?

9.- Table 12 translate to English

Author Response

We have corrected the text according to the reviewer's instructions. I am sending you the responses to the comments below.

The topic is of great interest. Unfortunately, graphs, tables appear in the paper without being properly explained in the writing or even without any reference to them in the text. There is no references to support the equations used to calculate the  figures of merit.     

Done

The following list try to highlight the main drawbacks that I have found to understand this work (P: page; L: line)

1.- P.9 L 325. Equivalence Statistical Test:  in this paragraph the procedure is not clear (do the authors apply Soxhlet method to some samples?). It seems that the equivalence test is passed, does not it?

Better explained in the paper. The statistical test was already done.

2.- P.9 L332: method Optimisation. I understand that samples 87, 88, 89, 89, 90 correspond to runs 1, 2, 3, 4 respectively. But, then, the column “Nº of Tests”  should be the same in tables 2,3,4,5, but it is not.

Done

As a matter of fact only Table 3 and 5 seem right attending to L. 338 -339 (“runs number 1 and 2 were analysed in triplicate and runs 3 and 4 in duplicate”)  which seem to say that 10 measurements should appear in each Table.

3.- Figures 4, 5, 6, 7 correspond to samples 1, 2, 3 4  in Table 2?.  If so, average line in the graph is not coherent with the average in Table 2.  Besides this, control charts are represented over time (L. 374) but the authors do not say that measurements are taken throughout time.

Corrected

4.-P.12 L.401 Validation study. These are the “figures of merit”, is there a relevant reference from which  the authors take the corresponding definitions that apply later (Lines:  421, 424, 435, 454, 458 461) ?

Corrected

5.- P.12 and 13 L. 406-418  As 2 blanks are included per run, Table 6 should have 20 values instead of 18 ?  Grubbs test only detect if the maximum or minimum value is an outlier under normal distribution of the measurements, is this the goal  here? What do the authors mean with “interval of 0.05% “ in L.414, what is the role of Table 9 in this study?

Better

6.- From P.15 L.476 to P.19 L. 486 tables and graphs appear  without any explanation appear. What is their meaning? What do they appear for? What the reader should conclude after reading and studying each of these tables?

 This was raw data and was deleted

Minor comments:

1.-P.10 L.364 some reference for criterion CV<10%? 

Corrected

2.- L 380, 383, 385, 386  LAS, LAI , LCS;LISàHLA, LLA, …. 

Corrected

3.- P. 13 L.421, 423, what   x0 and S0 are. 

Corrected

4.- L.426 de sample?   

Corrected

5.- L. 440 and L.449 Why these references are not at the end of the paper?  

Corrected

6.- P.14 L454 translate to English    

Corrected

7.- L.468 Figure 81?      

Corrected

8.- L. 474 matrixes?               

Corrected

9.- Table 12 translate to English    

Corrected

Reviewer 2 Report

  1. The presented article raises an interesting issue of a new method that can be used for fat determination, but is written very carelessness.
  2. There are many formal errors.I have listed a few of them below:
  • different fonts are used in the manuscript,
  • there are formal errors in the citation of the reference (there should only be numbers of references in square brackets, in the manuscript there are the authors' names together with the year and the reference number, for example lines: 74-75, 76-77, 81-82, 95-96, 97, 114, 128, 140, 148, 152-153, 156, 172, 175),
  • there is no reference in the manuscript to the position [16] and [24], which are included in the list references,
  • captions should be placed under the figures and not above them (e.g., Figures 1 and 2).
  1. Abbreviations should be explained the first time they are used (e.g., LOD, LOQ, MRC in Abstract, Texp<Tcrit on the Figure 3).
  2. The "Introduction" chapter does not show any news.The presented general information on fats is generally known.In addition, the descriptions of the methods for determining the fat content (Soxhlet and NIR methods) are too long and there is no novelty in these part of manuscript.While reading the "Introduction" chapter, I thought I was reading the review article. I suggest describe in this section the specification of the material used (general information about the olives, which is the fat content and what factors influence the fat content of olives).
  3. In the chapter “Material, methods and procedure”, there is insufficient description of the tested material. The abstract states that the fat content was determined in 12 olive samples, and the section "samples" states that the samples were subdivided for the three different workflows. However, some tables show the results for four samples. There is also no description of what was the reference sample.
  4. Why cyclohexane was used for extraction?What is the dielectric constant of this solvent?
  5. Figures 1 and 4-8 lack the axis captions, which makes it difficult to interpret the figures. Figure 81 it should be 8.
  6. The equation in line 458 is incomprehensible.
  7. In the section "Results and discussion" there is no reference to Tables 7-12, there is also no description of the results listed in these tables.
  8. The authors write, that they also used Near Infrared spectroscopy method, but in the discussion of the results I did not find any references to the obtained results.
  9. I am interested in the quality of the fat extracted from the olives, since the extraction was carried out at 125ËšC.

Author Response

We have corrected the text according to the reviewer's instructions. I am sending you the responses to the comments below.

  1. There are many formal errors.I have listed a few of them below:
  • different fonts are used in the manuscript,
  • there are formal errors in the citation of the reference (there should only be numbers of references in square brackets, in the manuscript there are the authors' names together with the year and the reference number, for example lines: 74-75, 76-77, 81-82, 95-96, 97, 114, 128, 140, 148, 152-153, 156, 172, 175),
  • there is no reference in the manuscript to the position [16] and [24], which are included in the list references,
  • captions should be placed under the figures and not above them (e.g., Figures 1 and 2).
  • Corrected
  1. Abbreviations should be explained the first time they are used (e.g., LOD, LOQ, MRC in Abstract, Texp<Tcrit on the Figure 3).  

Corrected

  1. The "Introduction" chapter does not show any news.The presented general information on fats is generally known.In addition, the descriptions of the methods for determining the fat content (Soxhlet and NIR methods) are too long and there is no novelty in these part of manuscript.While reading the "Introduction" chapter, I thought I was reading the review article. I suggest describe in this section the specification of the material used (general information about the olives, which is the fat content and what factors influence the fat content of olives).

The main goal of references in introduction was to assure some revision of the current methods, just to better align the purpose of the work.

  1. In the chapter “Material, methods and procedure”, there is insufficient description of the tested material. The abstract states that the fat content was determined in 12 olive samples, and the section "samples" states that the samples were subdivided for the three different workflows. However, some tables show the results for four samples. There is also no description of what was the reference sample.

Corrected and better explained

  1. Why cyclohexane was used for extraction? What is the dielectric constant of this solvent?

https://sites.google.com/site/miller00828/in/solvent-polarity-table  

 The values for relative polarity are normalized from measurements of solvent shifts of absorption spectra and were extracted from Christian Reichardt, Solvents and Solvent Effects in Organic Chemistry, Wiley-VCH Publishers, 3rd ed., 2003

Cyclohexane was chosen over other solvents, because it could fulfil some of the properties for a good and quantitative  fat extraction. It has a similar polarity (0.006 Relative polarity) to hexane (0.009 Relative polarity). Additionally a somewhat higher boiling point it would help in the manipulations of solvents and lost by evaporation, yielding better quantitative results. As the low polarity is very important for the fat extraction, this characteristic, will made the microwave absorption from this solvent quite inefficient. The existence in the solvent media of water will contribute for microwave absorption and therefore a more effective heating of the reaction vessel content.

  1. Figures 1 and 4-8 lack the axis captions, which makes it difficult to interpret the figures. Figure 81 it should be 8.

Corrected

  1. The equation in line 458 is incomprehensible. 

Corrected

  1. In the section "Results and discussion" there is no reference to Tables 7-12, there is also no description of the results listed in these tables.

This was raw data and was deleted

  1. The authors write, that they also used Near Infrared spectroscopy method, but in the discussion of the results I did not find any references to the obtained results.

It may be a misinterpretation of what is described in abstract. It was only pointed some of the advantages of NIR over other methods.

  1. I am interested in the quality of the fat extracted from the olives, since the extraction was carried out at 125ËšC.

So do we, but the main objective of this work is not to verify qualitative composition of extracted fat. The total fat parameter is a quantitative determination only. That remains for subsequent works.

Round 2

Reviewer 1 Report

The authors have followed most of the indications in the previous review.

Nevertheless, in page 16-17 Grobbs test's application should be  clarified. As a matter of fact the confidence interval that appears in Lines 3-5 seems out of place. Besides this, only one outlier can be detected with this test where the normality assumption should be fulfilled.

 Page 18, L.15-18: this reference should appear in the "references" section, should not it?

Author Response

All the itens were revised according to last report.

Reviewer 2 Report

The revised paper has a good layout. The statistical and mathematical methods are  a strong point of the manuscript. Figure and tables are well organized, the references are also complete.

Author Response

(The authors gave the same response as above.)
